# REWARD MODEL ENSEMBLES HELP MITIGATE OVEROPTIMIZATION

**Thomas Coste**[1], **Usman Anwar**[1], **Robert Kirk**[2], **David Krueger**[1]
[1]University of Cambridge, [2]University College London
`{tc628,ua237,dsk30}@cam.ac.uk`, `robert.kirk.20@ucl.ac.uk`

## ABSTRACT

Reinforcement learning from human feedback (RLHF) is a standard approach for fine-tuning large language models to follow instructions. As part of this process, learned reward models are used to approximately model human preferences. However, as imperfect representations of the "true" reward, these learned reward models are susceptible to *overoptimization*. Gao et al. (2023) studied this phenomenon in a synthetic human feedback setup with a significantly larger "gold" reward model acting as the true reward (instead of humans) and showed that overoptimization remains a persistent problem regardless of the size of the proxy reward model and training data used. Using a similar setup, we conduct a systematic study to evaluate the efficacy of using ensemble-based conservative optimization objectives, specifically worst-case optimization (WCO) and uncertainty-weighted optimization (UWO), for mitigating reward model overoptimization when using two optimization methods: (a) best-of-n sampling (BoN) (b) proximal policy optimization (PPO). We additionally extend the setup of Gao et al. (2023) to include 25% label noise to better mirror real-world conditions. Both with and without label noise, we find that conservative optimization practically eliminates overoptimization and improves performance by up to 70% for BoN sampling. For PPO, ensemble-based conservative optimization always reduces overoptimization and outperforms single reward model optimization. Moreover, combining it with a small KL penalty successfully prevents overoptimization at no performance cost. Overall, our results demonstrate that ensemble-based conservative optimization can effectively counter overoptimization.

## 1 INTRODUCTION

With the advent of large language models, reinforcement learning from human feedback (RLHF) has emerged as a powerful technique to fine-tune and enhance models' behaviors (Ziegler et al., 2019; Ouyang et al., 2022; Bai et al., 2022a). However, despite its empirical success, RLHF remains a fickle method suffering from many failure modes (Casper et al., 2023). One such failure mode is *overoptimization*, a phenomenon in which policy optimization appears to be making progress according to the learned reward model, but in reality begins to regress with respect to the true reward function (Ziegler et al., 2019; Stiennon et al., 2020; Gao et al., 2023). While many works on RLHF contain anecdotal evidence of overoptimization (Ziegler et al., 2019; Stiennon et al., 2020; Dubois et al., 2023), Gao et al. (2023) is the only work that studies overoptimization in a systematic way. As working directly with human labelers is expensive, Gao et al. (2023) introduce a synthetic setup to study overoptimization in which a much larger language model is first trained as a "gold" reward model and is then used to generate preference labels for training of proxy reward models.

In this work, we conduct a systematic study investigating whether combining ensembles with conservative optimization can help mitigate overoptimization. Our results indicate that not only does ensemble-based conservative optimization help mitigate overoptimization, it also results in improved performance. Our setup is identical to that of Gao et al. (2023) with one modification: the addition of label noise. Gao et al. assume that the preference labels used to train the proxy reward model do not contain any noise. However, this does not mirror the real-world RLHF setup, in which agreement rates among human annotators are typically between $60 - 75\%$ (Ziegler et al.,

---

The code is available at: `https://github.com/tlc4418/llm_optimization`.

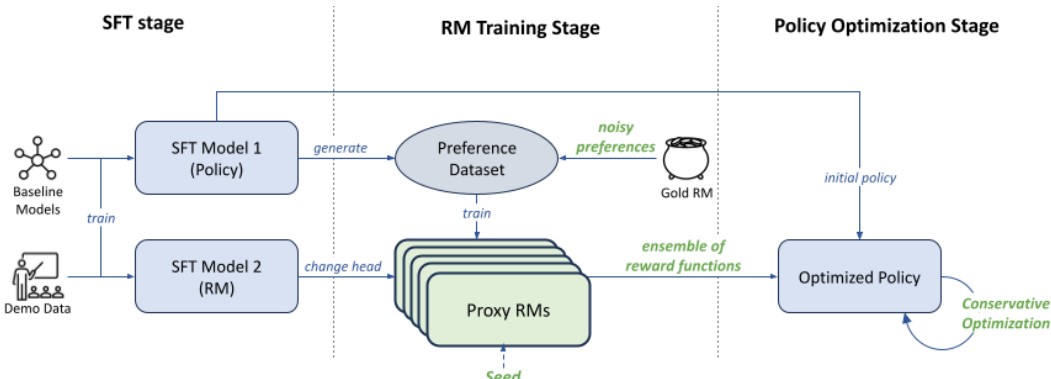

Figure 1: RLHF pipeline used in this work - our modifications on top of the standard RLHF setup used in Gao et al. (2023) are highlighted in green.

2019; Stiennon et al., 2020; Dubois et al., 2023). To simulate that disagreement and to better reflect the real-world RLHF, we extend the setup to include 25% label noise as well. In both the cases of no label noise and 25% label noise, we provide strong evidence that ensemble-based conservative optimization methods are effective in mitigating overoptimization and improving performance.

Scaling laws for reward model overoptimization discovered by Gao et al. (2023) indicate that increasing the size of the proxy reward model reduces overoptimization as well. However, reward models are derived from pretrained language models. Thus, acquiring a larger reward model would require significant *pretraining*, which is not always feasible and can be very costly (Morgan, 2022; Venigalla & Li, 2022). However, our approach, using ensembles of reward models, only requires fine-tuning multiple copies of an already pretrained reward model, which is relatively inexpensive. Moreover, our model and data scaling results (Figures 8 and 9) indicate that the gains provided by our method are orthogonal to the gains achieved by increasing the reward model size; thus, the two approaches can be combined seamlessly for even better results.

Our main contributions are as follows:

- We conduct the first study of using ensembles to counter overoptimization in RLHF-based fine-tuning of language models.
- Our results indicate that using ensembles and conservative optimization eliminates overoptimization for BoN and results in up to 70% improvement in some cases.
- For PPO, ensemble-based conservative optimization typically outperforms single reward model optimization, and when combined with a suitable KL penalty weight successfully eliminates overoptimization.
- We further conduct studies to establish the robustness of the ensemble-based conservative optimization methods to any new hyperparameters it introduces (e.g. size of the ensemble).

## 2 BACKGROUND

In this section, we briefly review two commonly used policy optimization methods: best-of-n sampling (BoN) and proximal policy optimization (PPO), followed by a discussion of overoptimization.

### 2.1 BEST-OF-N SAMPLING (BoN)

Best-of-$n$ (BoN) sampling, also called rejection sampling, is a simple inference-time optimization method (Ouyang et al., 2022; Nakano et al., 2021). For a given prompt, $n$ responses are generated from the policy model, and the answer with the highest proxy reward model score is returned. To evaluate the degree of optimization, the KL distance is defined analytically as a function of $n$:

$$\text{KL}_{\text{bon}} = \log n - \frac{n-1}{n} \qquad (1)$$

## 2.2 Proximal Policy Optimization (PPO)

Proximal Policy Optimization (Schulman et al., 2017) is a policy-gradient-based online reinforcement learning method that maximizes a given reward function by repeatedly performing small incremental updates to the policy. PPO is the standard algorithm used in fine-tuning language models based on human feedback (Ouyang et al., 2022; Bai et al., 2022a; Stiennon et al., 2020; Zheng et al., 2023). When using PPO to fine-tune a language model, a KL penalty term is added during the reward calculation to regularize the policy by preventing it from deviating far from the initial policy:

$$R^{\text{PPO}}(q, a) = R(q, a) - \beta \log \left[ \frac{\pi^{\text{PPO}}(a|q)}{\pi^{\text{init}}(a|q)} \right] \tag{2}$$

where $\pi^{\text{PPO}}$ is the policy being optimized and $\pi^{\text{init}}$ is the initial (pretrained) language model. The degree of optimization is measured in terms of KL distance between the initial policy and the one being optimized, with details for this calculation provided in Appendix C.

## 2.3 Overoptimization

In reinforcement learning from human feedback (RLHF) a reward model is used to approximate human preferences, to remove the need for querying humans for every policy generation. As the learned reward model is only a proxy for the true reward function, optimizing it may not always result in an improvement according to true human preferences. In practice, optimizing a (fixed) learned reward model almost always leads to improvement according to this learned reward model, but only improves according to the *true* reward model (i.e. humans) for some initial period, after which performance often begins to regress. This phenomenon is referred to as *overoptimization*, an example of which is shown in Figure 2.

To study the problem of *overoptimization*, Gao et al. (2023) introduced a synthetic setup in which, instead of human annotators, a *gold* reward model is used to score responses and generate preferences to train proxy reward models. The gold reward model is generally chosen to be much larger than the proxy reward model, to simulate the fact that in a real setup, human preferences are too complex to be captured by a neural network with a finite capacity. Within this setup, Gao et al. (2023) discover overoptimization to be a persistent issue, although they note that larger proxy reward model sizes and greater amounts of training data can help reduce overoptimization. However, scaling up is not always a feasible solution as the proxy reward models are generally derived from extensively pretrained language models.

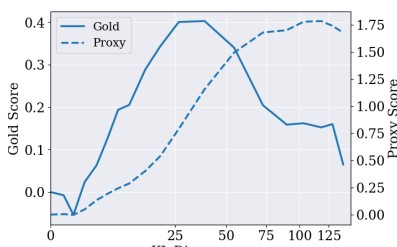

Figure 2: An example of overoptimization. The KL divergence is the distance between the initial and current policy, measuring the degree of optimization.

## 3 Method

Standard RLHF often learns a single reward model to estimate the true reward, which is then used to optimize the policy. However, many works in wider machine learning have shown that learning multiple estimators and combining them can help improve robustness (Lakshminarayanan et al., 2017; Ovadia et al., 2019; Yu et al., 2020b). Taking inspiration from this insight, we propose to learn an ensemble of reward models $\{R_1, ..., R_k\}$ in the reward model training stage. During policy optimization, we combine the reward estimates from different reward models within the ensemble according to the following three methods.

**Mean Optimization**: Mean optimization (Boyd & Vandenberghe, 2004; Wright, 2006) simply takes the mean of the outputs of the different ensemble members:

$$R_\mu(q, a) := \frac{1}{k} \sum_{i=1}^{k} R_i(q, a) \tag{3}$$

where $q$ is the prompt given to language model and $a$ is the corresponding response sampled from the language model.

We note that mean optimization is not a conservative optimization technique; if at least one member of the ensemble overestimates the reward (even while other members accurately estimate it), mean optimization will not prevent the policy from exploiting the faulty reward model.

**Worst-Case Optimization**: Worst-case optimization (WCO) (Boyd & Vandenberghe, 2004; Wright, 2006) creates a conservative estimate by choosing the lowest reward from the ensemble at every step. Choosing the lowest reward helps ensure that as long as at least one ensemble member does not overestimate the true reward, policy optimization will not result in overoptimization.

$$R_{\text{WCO}}(q, a) := \min_i R_i(q, a) \tag{4}$$

A major advantage of WCO is that it does not have any hyperparameters that might require tuning. However, it can sometimes result in a performance penalty due to its highly conservative nature.

**Uncertainty-Weighted Optimization**: In uncertainty-weighted optimization (UWO) (Wu et al., 2021b; Yu et al., 2020b; Brantley et al., 2019), the reward for a sample is calculated by combining the average reward across all models in an ensemble with the intra-ensemble variance, weighted by a coefficient $\lambda$. Intuitively, UWO works by penalizing the policy for generating responses for which there is high disagreement among reward models within the ensemble. This helps prevent the exploitation of a single faulty reward model which might be erroneously assigning high rewards to incorrect or low-quality responses. Mathematically, this objective is given by:

$$R_{\text{UWO}}(q, a) := \underbrace{\frac{1}{k} \sum_i R_i(q, a)}_{\text{mean}} - \lambda \underbrace{\frac{1}{k} \sum_i \left( R_i(q, a) - \frac{1}{k} \sum_i R_i(q, a) \right)^2}_{\text{variance}} \tag{5}$$

where $\lambda$ is a hyperparameter which controls the weight of the uncertainty component.

## 4 EXPERIMENTAL SETUP

We based our experiments on the setup proposed by Gao et al. (2023), however, we use open source models (Biderman et al., 2023; Dubois et al., 2023), open source datasets (Taori et al., 2023), and train our proxy reward models under $25\%$ label noise. We also present qualitative reproduction of the results of Gao et al. in Appendix E.

### 4.1 DATA

In order to train proxy reward models, we use the Alpaca dataset (Taori et al., 2023), with $52,000$ instructions covering a range of commands and corresponding demonstrations generated by OpenAI's text-davinci-003 (OpenAI, 2023a). Each entry is composed of an instruction, an optional additional input, and a demonstrator output. More specifically, we use the AlpacaFarm (Dubois et al., 2023) variant of the dataset, as it provides splits for use in the different RLHF stages and for validation, as well as human preference annotations. Further details concerning the splits, prompt format, and examples are given in Appendix D.

### 4.2 PRETRAINED MODELS

For the policy model and (proxy) reward model, we use pretrained language models provided in the Pythia suite (Biderman et al., 2023). The policy model used everywhere in this work is the 1.4B Pythia model. For proxy reward models, we remove the unembedding layers from Pythia models of sizes 14M, 70M, and 1.4B and add a scalar head to output the reward to get proxy reward models of sizes 7M, 44M, and 1.3B.

Finally, the 7B human preference reward model from AlpacaFarm (Dubois et al., 2023) is used as the gold reward model. It is of very similar size to the (closed-source) 6.9B gold reward model used in Gao et al. (2023) and is significantly larger than any of our proxy RMs (with the largest being 1.3B) and as such is a reasonable choice for a *gold* standard.

### 4.3 RLHF Pipeline

Our RLHF pipeline is similar to that of Gao et al. (2023) with several modifications highlighted in Figure 1. We explain the full setup below for completeness.

**Supervised Fine-tuning**: As the first step in our RLHF pipeline, both the policy model and the proxy reward model undergo supervised fine-tuning using 10k instruction demonstrations from the "sft" split of the AlpacaFarm dataset (see Section 4.1 and Appendix D.2 for details). This fine-tunes the models to have better instruction-following capabilities.

**Preference Generation and Labeling**: To generate a preference dataset, the SFT model is prompted using instructions from the AlpacaFarm dataset, for which it produces two responses per instruction. Relevant hyperparameters for sampling these responses are given in Appendix D.1. The two responses are then scored with the gold reward model, which assigns a score to each of them. Human annotators tend to have high disagreement rates, often around $25\%$ (Stiennon et al., 2020) or more (Ziegler et al., 2019; Dubois et al., 2023). To simulate this, we optionally mislabel $25\%$ of the dataset. This results in two datasets: one with no label noise and one with $25\%$ label noise.

**Proxy Reward Model Training**: We train our proxy reward model by minimizing cross-entropy loss on the preference dataset generated in the previous timestep. Unless mentioned otherwise, we use the complete dataset of $46,000$ prompts for reward model training and train all the reward models for 5 epochs. We give other hyperparameters for reward model training in Appendix D.1 and example validation loss curves for a set of reward models in Appendix F.1. Trained reward models reach between 60-75% validation accuracy.

**Ensemble Creation**: To create an ensemble, we train a fixed number of proxy reward models using identical data and hyperparameters but with different random seeds. This results in different random initialization for the scalar reward head added on top of the pretrained language model and a different data shuffling order. We use all the available training data for the training of every reward model as training on lesser data results in higher validation loss and poorer performance (Lakshminarayanan et al., 2017). Unless stated otherwise, we train an ensemble consisting of five reward models.

**Policy Optimization**: Similar to Gao et al. (2023), we use BoN sampling and PPO as optimization algorithms. For BoN, the evaluation cost for greater KL nats increases exponentially. As a result, due to constraints on available compute, we only evaluate BoN for a maximum of $n_{max} = 12,500$ samples[1], which roughly equals $8.4$ nats of KL. For PPO, we train for 3000 PPO steps. We give further details on implementation and other hyperparameters in Appendix D.1.

## 5 Results

In this section, we report the results of our experiments. To summarize, our main findings are:

- Using an ensemble, with any of the objectives highlighted in Section 3, almost always helps over using a single reward model.
- For BoN, we do not observe any overoptimization when using WCO and UWO as optimization methods; however, in the $25\%$ label noise case, mean optimization does overoptimize (Figure 3).
- For BoN, all three ensemble-based optimization methods result in up to $\sim 30\%$ improvement in final performance over the average performance attained by optimizing individual reward models in the no label noise case and up to $\sim 75\%$ improvement in the $25\%$ label noise case (Figure 3).
- For PPO, WCO, and UWO do reduce overoptimization (Figure 5), but completely mitigating overoptimization requires combining them with a small KL penalty (Figure 4).
- For PPO, WCO, and UWO always match or outperform single reward model optimization in terms of final performance for all values of KL penalties (Figure 7).
- Our findings are robust to scaling of model size and training dataset size (Figures 8 and 9).

To present our results, we primarily rely on policy optimization curves where we show the gold reward model score on the left y-axis and the proxy reward model score on the right y-axis. For

---

[1]Generating the $n = 12,500$ answers for 1000 prompts and then relabeling them with proxy and gold reward model takes approximately 700 A100 GPU hours.

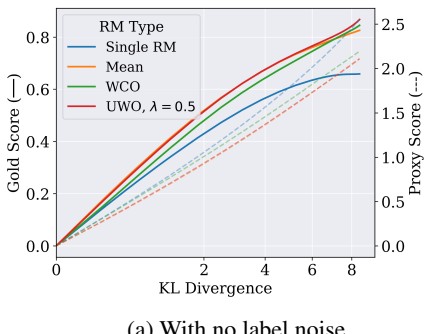 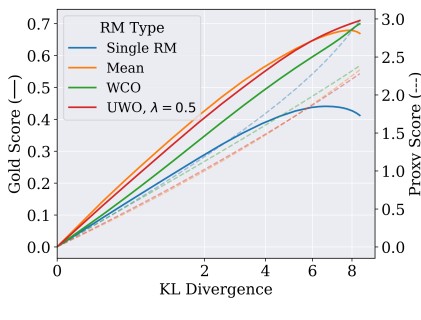

(a) With no label noise.

(b) With 25% label noise.

Figure 3: BoN results for 44M reward model size. In this and future BoN Figures, KL divergence is defined as per Eq. 1 and measures the degree of optimization.

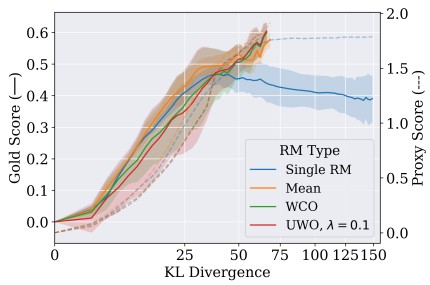 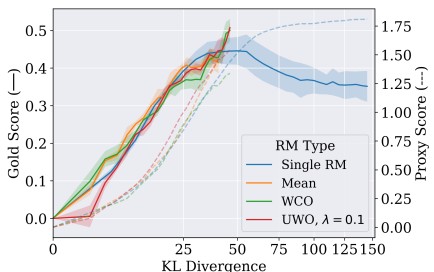

(a) With no label noise. For all four methods, KL penalty 0.01 is used.

(b) With 25% label noise. KL penalty 0.1 for mean and 0.01 for all other methods is used.

Figure 4: PPO results for 44M reward model size, over three PPO seeds. The KL divergence scale is different from BoN due to differences in the algorithm and the KL calculation (Appendix C). For each method, we choose the KL penalty that gives the best final performance.

the x-axis, following the common practice (Bai et al., 2022a; Gao et al., 2023), we use the KL divergence as defined in Eq. 1 or Appendix C, on a square root scale for clarity. For PPO, most runs do not exceed a KL divergence of 150, but those that do are truncated to maintain a more legible plot. Additionally, runs in Figure 4, the main PPO optimization illustration, are averaged over three PPO seeds, with standard deviation shown. For other PPO optimization figures, due to the low variance observed, a single seed is used. However, the results for single RMs are averaged over the five RMs which make up the ensemble for the other methods. When comparing many experiment results at once, we present bar plots showing the final performance on the gold reward model and the variable of interest on the x-axis. As an additional evaluation metric for quality, Appendix F.9 compares the final policy win-rate of different ensemble methods. Finally, we also include some qualitative samples in Appendix F.10.

## 5.1 BEST-OF-N SAMPLING

In Figure 3, we present results for BoN sampling for a 44M size reward model. Across both noiseless and noisy settings, ensembles help improve performance by $\sim 30\%$ and $\sim 75\%$ respectively and successfully avoid overoptimization, except for mean optimization in the case of noisy labels. For UWO, we show results for $\lambda = 0.5$ which we found to be most performant; however, in Section 5.4, we show that many reasonable values of $\lambda$ work well. Additional results for different sizes of reward models are presented in Appendices F.3 and F.4.

## 5.2 PROXIMAL POLICY OPTIMIZATION

For PPO we observe that WCO and UWO reduce overoptimization and outperform other methods in terms of final performance in the absence of KL penalty, although they do not *eliminate* overoptimization completely (as shown in Figure 5). However, in Figure 4, we show that with a small KL

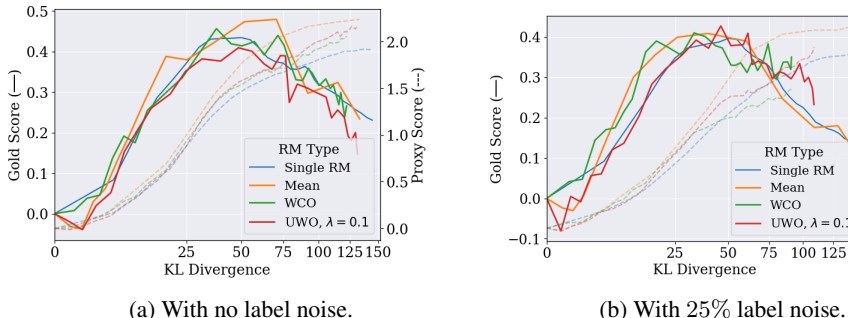

(a) With no label noise.
(b) With 25% label noise.

Figure 5: PPO results for different optimization objectives without using a KL penalty.

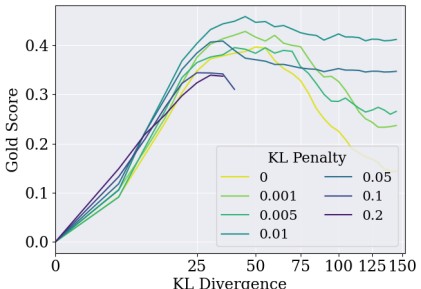

Figure 6: Individual reward model optimization (i.e. no ensemble) for different values of KL penalty in the 25% label noise case.

Figure 7: Effectiveness of conservative optimization methods across KL penalty weights in the 25% label noise case.

penalty coefficient of 0.01, WCO and UWO both successfully prevent overoptimization, without any notable reduction in performance. On the other hand, when using KL penalty on its own, a 20x larger KL penalty weight of 0.2 is required to eliminate overoptimization which incurs a significant performance penalty (Figure 6). In Figure 7, we show that for all penalties, WCO and UWO match or outperform the average final performance achieved by optimizing a single reward model. Moreover, for small KL penalties, they comprehensively outperform single reward model optimization.

## 5.3 MODEL AND DATA SCALING RESULTS

Prior work (Gao et al., 2023) has shown that increasing reward model size or training dataset size helps improve performance as measured by the gold reward model. In Figure 8, we vary the size of reward models (keeping the dataset size fixed at 46k samples and policy size fixed at 1.4B) and plot the final performance of each method for the fixed training budget. Recall that for BoN, this is $n = 12,500$ samples and for PPO, this is 3000 timesteps. Our results show that the improvement due to the use of ensembles is orthogonal to the improvement in performance achieved by scaling the reward model size. In particular, with the 1.3B reward model, we highlight that even when the policy and reward model have a similar number of parameters, WCO and UWO still provide non-trivial gains over using a single reward model. A more detailed illustration of this example can be found in Appendix F.4.

Similarly, in Figure 9, we vary the size of the training dataset (keeping the reward model size fixed at 44M) and plot the final performance for each method for the fixed training budget. The results again indicate that using ensembles provides additional improvement in addition to increasing the size of the training dataset.

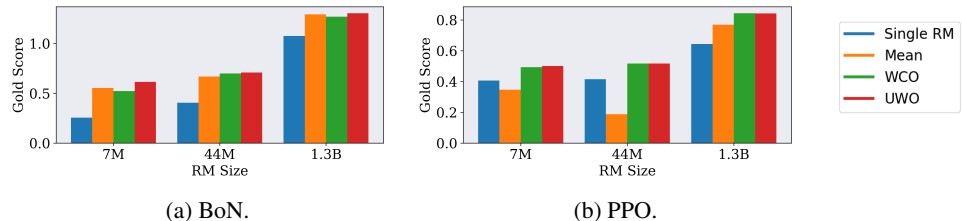

(a) BoN.    (b) PPO.

Figure 8: Final gold reward model performance achieved by different objectives when optimizing reward models with varying parameter sizes, but trained with the same dataset containing 25% label noise. The 1.3B PPO models are exceptionally trained for 6000 PPO steps, due to a slower policy optimization rate with large reward models (Appendix F.4).

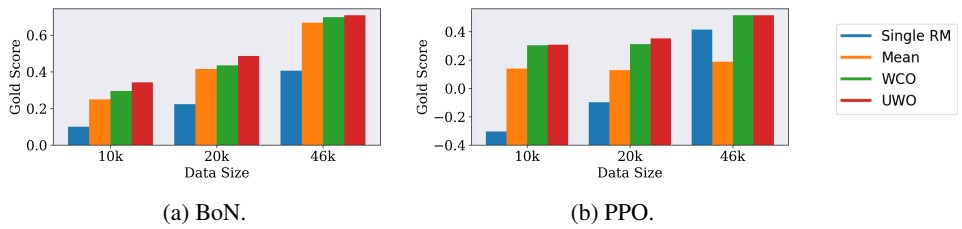

(a) BoN.    (b) PPO.

Figure 9: Final gold reward model performance achieved by different objectives when optimizing (44M) reward models trained under varying amounts of data. A label noise ratio of 25% was maintained for all dataset sizes.

## 5.4 Evaluating Robustness to Hyperparameters

Using an ensemble-based approach introduces an additional hyperparameter: the *cardinality*, or size, of the ensemble. In Figure 11, we plot the final performance for BoN and PPO for the three ensemble-based approaches: mean optimization, WCO, and UWO. We note that while there is a noticeable gap between 3-member and 4-member ensembles, in most cases the performance is highly similar for 4-member and 5-member ensembles, indicating that 4-member or 5-member ensembles are likely to work best, and diminishing returns will be seen after this point.

For UWO, the value of the uncertainty penalty is another hyperparameter. Our results in Figure 12 indicate that most reasonable values of uncertainty penalty actually work well, indicating that there is potentially no need to tune this hyperparameter.

## 5.5 Effects of Label Noise and Uncertainty Penalty

In Figure 10, we show intra-ensemble variance for an ensemble of 44M parameter reward models optimized via UWO and mean optimization objectives using PPO. The variance among the ensemble members starts at a small value in the no label noise case, and increases by a relatively small amount during training for UWO. However, the uncertainty increases by almost 3 times for mean optimization during training.

For the case of 25% label noise, the variance starts much higher and during the course of training, increases by about 2.5 times for mean optimization. However, the variance only increases by about 20% for UWO (using $\lambda = 0.1$). Further, using a KL penalty of 0.01 results in a slight reduction in the variance at the

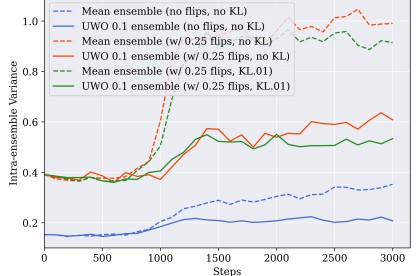

Figure 10: Intra ensemble variance for different optimization objectives under different levels of noise for PPO.

end of training. This hints at the fact that while mean optimization is able to exploit any reward model that overestimates the reward, the uncertainty penalty in UWO prevents that - resulting in better final performance (shown in Figure 7) and reduced overoptimization.

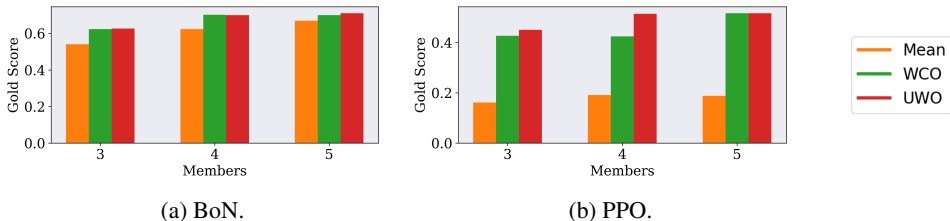

(a) BoN.      (b) PPO.

Figure 11: Impact of the cardinality of the ensemble on the final performance of mean, UWO, and WCO for the $25\%$ label noise case.

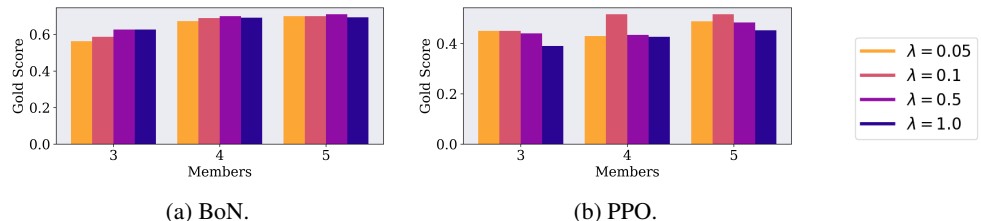

(a) BoN.      (b) PPO.

Figure 12: Impact of uncertainty penalty weight on final performance for different numbers of reward models within the ensemble. We note that there is considerable robustness to the value of the uncertainty penalty.

## 6 DISCUSSION

We have demonstrated that ensemble-based conservative optimization methods improve performance and are highly effective in combating the overoptimization problem in RLHF. This work opens up the possibility of several interesting follow-ups. Firstly, future work should consider replicating our findings in other environments (i.e. other RLHF datasets) and with larger-scale language models. Secondly, our setup is based on *offline* RLHF, in which human feedback is collected upfront and there are no updates made to the reward model throughout the policy optimization process. This contrasts with online RLHF (Bai et al., 2022a; Christiano et al., 2017), where reward models are periodically retrained on freshly collected data from humans. Does ensembling result in similar gains in this setup as well or not?

## 7 RELATED WORKS

**Overoptimization in RLHF**: Several works (Ibarz et al., 2018; Ziegler et al., 2019; Stiennon et al., 2020) have provided anecdotal evidence of overoptimization in RLHF. However, to the best of our knowledge, Gao et al. (2023) are the only ones who study it systematically within the setup of fine-tuning of LLMs. Our work utilizes their setup, reproduces their results, and performs a systematic study evaluating the effectiveness of using ensembles for mitigating overoptimization.

**Use of Ensembles in (Model-Based) RL**: Ensembles are often used in deep learning as a tool to estimate uncertainty (Lakshminarayanan et al., 2017; Dietterich, 2000) and can often outperform more complex Bayesian deep learning methods (Ovadia et al., 2019). Learning an ensemble of dynamics models is especially popular in model-based reinforcement learning where often a reliable uncertainty estimate over state space is critical for avoiding distribution shift (Depeweg et al., 2016; Gal et al., 2016; Chua et al., 2018; Yu et al., 2020a). Disagreement among ensemble members is also a popular approach for driving the exploration of learning agents in an environment (Henaff, 2019; Pathak et al., 2019; Shyam et al., 2019; Lee et al., 2021). Brantley et al. (2019) uses an ensemble-based approach to estimate the support of expert policies in an imitation learning setup. Within language models setting, Gleave & Irving (2022) explore the use of ensembles for uncertainty-based active learning to improve sample efficiency of reward models. However, to the best of our knowledge, there is no prior work that explores the use of ensembles for improving the robustness of RLHF; especially in the setting of language models fine-tuning.

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

APPENDIX

## A  ADDITIONAL RELATED WORKS

**RLHF**: Reinforcement learning from human feedback (RLHF), in its modern form, was popularized by Christiano et al. (2017). Ziegler et al. (2019) was the first work to demonstrate the effectiveness of RLHF for fine-tuning language models. This led to several applications of RLHF to fine-tuning LLMs focused on improving several aspects of language models e.g. instruction following (Ouyang et al., 2022; Abramson et al., 2022), summarization capabilities (Stiennon et al., 2020; Wu et al., 2021a), web browsing (Nakano et al., 2021), translation (Gulcehre et al., 2023) and helpfulness & harmlessness (Bai et al., 2022a). This has also caused a great amount of interest in improving RLHF as a method. One main line of research here is methods that try to augment the learning signal for the reward model in some form e.g. language feedback (Scheurer et al., 2023) or process supervision (Lightman et al., 2023; Uesato et al., 2022). This, however, generally incurs additional costs in labeling. This has prompted research on extracting feedback signal from a pretrained language model via appropriate prompting (Bai et al., 2022b; OpenAI, 2023b; Madaan et al., 2023; Saunders et al., 2022), however, this requires effective prompting and a highly capable language model to be successful. Another line of research focuses on the use of alternative algorithms to PPO for policy optimization step (Gulcehre et al., 2023; Rafailov et al., 2023; Zhao et al., 2023; Yuan et al., 2023; Dong et al., 2023; Xu et al., 2023). Orthogonal to both these lines of work, we explore using ensembles to improve RLHF.

**Reward Hacking**: Overotpimization in RLHF is an instance of reward hacking. Many examples of reward hacking exist in literature (Clark & Amodei, 2016; Lehman et al., 2020; Krakovna et al., 2020). Skalse et al. (2022) provide a formal definition for reward hacking and theoretically study whether unhackable proxies can exist or not under different conditions. Zhuang & Hadfield-Menell (2020) study the reward hacking that might arise due to partial observability. Pan et al. (2022) provide examples of proxy reward functions in various reinforcement learning environments where low-capacity policies do not elicit reward hacking but policies with greater capacity do.

## B  BON ESTIMATOR

The naive way of estimating the average reward of the BoN policy under a gold reward model, when using a proxy reward model as the ranking function, is to use a Monte Carlo estimate in which $n$ answers $\{A_1, ..., A_2\}$ are sampled from a language model $f$ and then scored for a given prompt $q$ as follows:

$$R_n^{\text{gold}}(q) := \mathbb{E}_{A_1, A_2, ..., A_n \sim f(q)} \left[ R^{\text{gold}} \left( \underset{a \in \{A_1, A_2, ..., A_n\}}{\arg\max} R^{\text{proxy}}(a|q)|q \right) \right] \quad (6)$$

This is very wasteful as it does not reuse answers for different values of $n$. Therefore, we instead use the following unbiased estimator (Nakano et al., 2021) that samples $N \geq n_{max}$ outputs for a given input $q$:

$$R_n^{\text{gold}}(q) = \frac{1}{\binom{N}{n}} \sum_{1 \leq i_1 < \cdots < i_n \leq N} R^{\text{gold}} \left( \underset{a \in \{A_{i_1}, ..., A_{i_n}\}}{\arg\max} R^{\text{proxy}}(a \mid q) \mid q \right) \quad (7)$$

This can be computed efficiently by first sorting all answers $A_1, ..., A_N$ under the proxy reward model to obtain scores $S_1, ..., S_N$ and then computing:

$$\frac{1}{\binom{N}{n}} \sum_{1 \leq i_1 < \cdots < i_n \leq N} R^{\text{gold}} \left( \underset{a \in \{S_{i_1}, ..., S_{i_n}\}}{\arg\max} R^{\text{proxy}}(a \mid q) \mid q \right) = \sum_{i=n}^{N} \frac{\binom{i-1}{n-1}}{\binom{N}{n}} R^{\text{gold}}(S_i \mid q) \quad (8)$$

To get the gold reward model score for a given range of values $n = 1, 2, ..., n_{max}$, we simply evaluate the empirical average of the above estimator for all questions for each $n$.

## C  KL DISTANCE CALCULATION FOR PPO

The naive way to calculate KL distance between the PPO-optimized policy $\pi_{\text{RL}}$ and the pretrained model is as follows:

$$\text{KL}_{\text{RL}}(\pi_{\text{RL}}, \pi_{\text{init}}) = E_{x \sim \pi_{\text{RL}}} \left[ \log \frac{\pi_{\text{RL}}}{\pi_{\text{init}}} \right] \tag{9}$$

However, this estimator has high variance and can be negative, unlike actual KL. Therefore, we use the following estimator (Schulman, 2020):

$$\text{KL}_{\text{RL}}(\pi_{\text{RL}}, \pi_{\text{init}}) = E_{x \sim \pi_{\text{RL}}} \left[ \frac{1}{2} \left( \log \frac{\pi_{\text{RL}}}{\pi_{\text{init}}} \right)^2 \right] \tag{10}$$

## D  ADDITIONAL EXPERIMENTAL DETAILS

### D.1  HYPERPARAMETERS

We give the hyperparameters here for different components of our RLHF pipeline:

Table 1: SFT hyperparameters.

| Parameter | Value |
|---|---|
| Learning rate | 8e-6 |
| Epochs | 3 |
| Batch size | 4 |

Table 2: RM hyperparameters.

| Parameter | Value |
|---|---|
| Learning rate | 1e-5 |
| Epochs | 5 |
| Batch size | 32 |

Table 3: PPO hyperparameters.

| Parameter | Value |
|---|---|
| Learning rate | 1e-6 |
| Cosine annealing scheduler | 1e-7 |
| PPO epochs | 4 |
| Batch size | 32 |
| Number of rollouts | 256 |
| Chunk size | 32 |
| Clipping range & value | 0.2 |
| GAE lambda | 0.95 |

### D.2  ALPACAFARM DATASET DETAILS

The AlpacaFarm dataset (Dubois et al., 2023) employed in our experiments uses the Alpaca data (Taori et al., 2023) made up of 52,000 samples. This data is chosen due to its large size and success in training instruction-following models. AlpacaFarm contains five splits: a labeled 10k "sft" split for

Table 4: Generation hyperparameters.

| Parameter | Value |
|---|---|
| Max instruction length | 520 |
| Max new tokens (answer length) | 256 |
| PPO epochs | 4 |
| Top-p | 0.9 (1.0 for PPO training) |
| Top-k | 0 |
| Temperature | 1.0 |

Table 5: Example answer generation and reward modeling prompts with proper formatting.

| Answer generation prompt | Reward modelling prompt |
|---|---|
| `<\|prompter\|>What geometric shape has 5 sides and 5 angles?<\|endoftext\|> <\|assistant\|>` | `<\|prompter\|>What geometric shape has 5 sides and 5 angles?<\|endoftext\|> <\|assistant\|>The geometric shape is a pentagon.<\|endoftext\|>` |
| `<\|prompter\|>Analyze the following sentence and identify the verb and its tense.\nShe played the piano.<\|endoftext\|> <\|assistant\|>` | `<\|prompter\|>Analyze the following sentence and identify the verb and its tense.\nShe played the piano.<\|endoftext\|> <\|assistant\|>Verb: Played Tense: Past<\|endoftext\|>` |

supervised fine-tuning, a 10k "pref" split containing pairwise preference labels, a 20k "unlabeled" split for training algorithms such as PPO, a 2k validation split, and an unused 10k split.

### D.3 PROMPT FORMAT

Though we use instructions from the AlpacaFarm dataset, this only provides content for prompting the LLM and still requires formatting. We opt for minimalism, following the v2 format used in OpenAssistant (Köpf et al., 2023). This format follows the `GPTNeoXTokenizer` class used to pretrain our LLMs and introduces two special tokens: `<|prompter|>` and `<|assistant|>`.

For answer generation, the model should be prompted with the instruction and the input. Inputs, should they be present, are appended to the instruction after a new line to form the prompt. The prompt is then prepended with the `<|prompter|>` token and closed off with an end-of-text (EOT) `<|endoftext|>` token declaring the end of the instruction, and the `<|assistant|>` token to start the answer. An example is shown in Table 5.

For reward modeling, the prompt should also contain an answer to be evaluated. In this case, the answer text (from an AlpacaFarm dataset demonstration or a previous answer generation) is appended to the initial prompt containing the instruction and closed off with the EOT token. This forms the full RM prompt, with an example shown in Table 5.

# E  REPRODUCTION OF RESULTS OF GAO ET AL. (2023)

We also successfully reproduced the results of Gao et al. in our setup shown in Figures 13 and 14. Note that our setup uses open-source models derived from the Pythia suite (Biderman et al., 2023) as the base for policy and proxy reward models and the AlpacaFarm human-preference reward model (Dubois et al., 2023) is used as the gold reward model. In contrast, Gao et al. used closed-source models based on GPT series. Our successful reproduction of their results in our setup hints at the general nature of overoptimization.

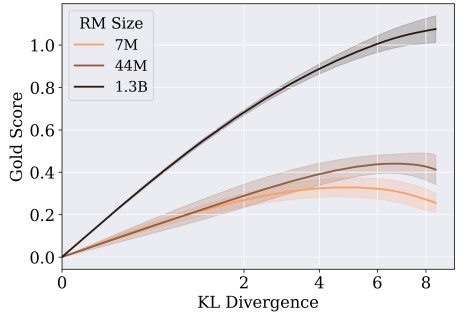

(a) BoN results for various reward model sizes, with data size held constant (46k).

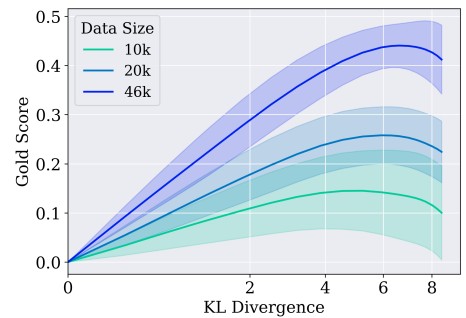

(b) BoN results for various reward model training data sizes, with RM size held constant (44M).

Figure 13: Gold reward model scores for BoN follow a similar trend as that observed by Gao et al. (2023) when varying reward model and data sizes. Runs are averaged over five seeds, with standard deviation additionally shown. 25% label noise is used here to highlight overoptimization, as we found it difficult to observe with our limited BoN optimization capability. This is in line with label noise ablation experiments from Dubois et al. (2023).

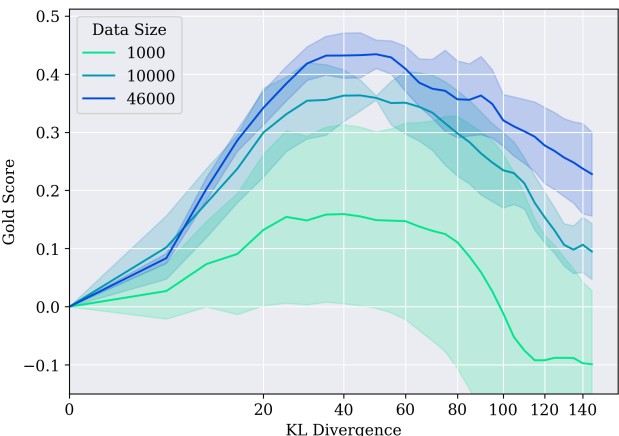

Figure 14: PPO optimization results for various reward model training data sizes, with reward model size held constant (44M). Average and standard deviation over three runs are shown. A similar trend to Gao et al. (2023) is observed, with greater overoptimization as optimization is pushed further.

# F ADDITIONAL RESULTS

## F.1 RM VALIDATION LOSS CURVE

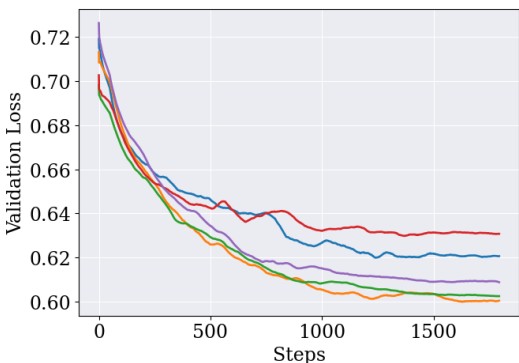

Figure 15: Reward model validation loss during training for 44M reward models with no label noise. Reward models are trained for 5 epochs, with each epoch lasting 359 steps. We note that while there is a minor variation in validation loss, this variation is not predictive of whether the reward model is robust to overoptimization or not. For example, in Figure 16, we see that the reward model with the highest validation loss does not suffer overoptimization but other reward models with lower validation losses do.

## F.2 INDIVIDUAL REWARD MODEL TRAINING OPTIMIZATION PLOTS

For the single reward model results, we optimize each reward model separately and present their average in Figures 3 and 4. Here, we present the training curve for each reward model separately.

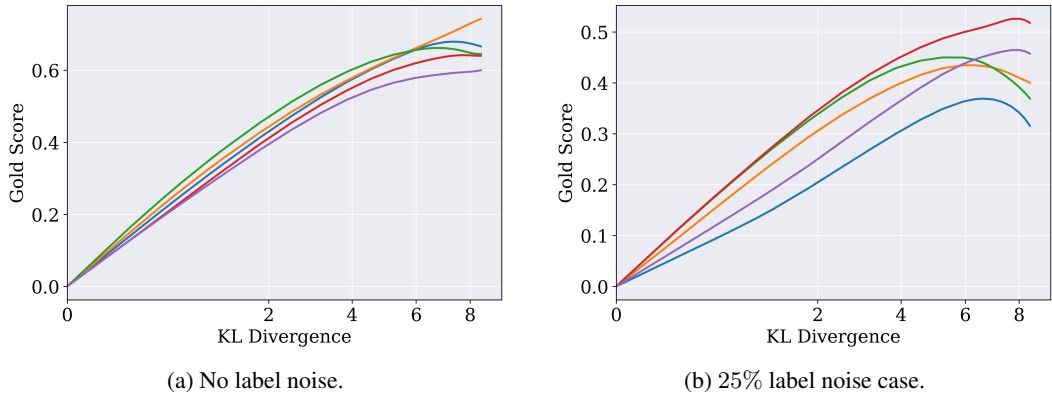

(a) No label noise.

(b) 25% label noise case.

Figure 16: BoN for 44M reward model. The average of these curves is presented in Figure 3.

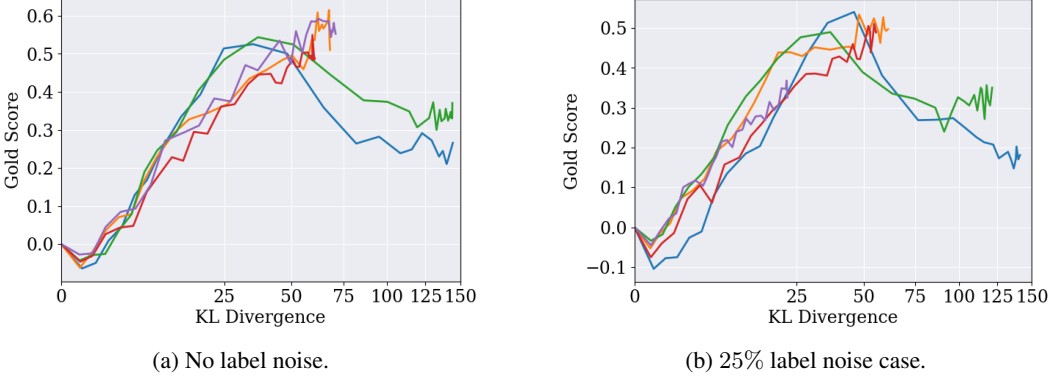

(a) No label noise.

(b) 25% label noise case.

Figure 17: PPO results for 44M reward model. The average of these curves is presented in Figure 4.

### F.3 7M REWARD MODEL TRAINING OPTIMIZATION RESULTS

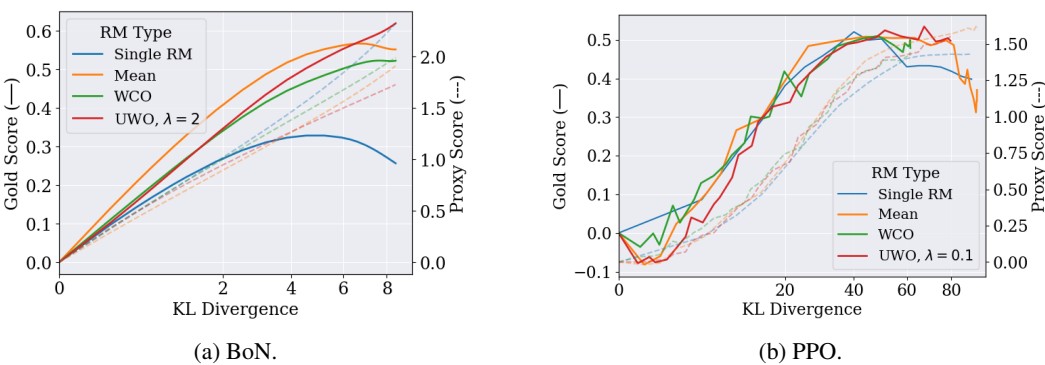

(a) BoN.       (b) PPO.

Figure 18: BoN and PPO results with the 7M reward models to supplement the bar plot in Figure 8.

### F.4 1.3B REWARD MODEL TRAINING OPTIMIZATION RESULTS

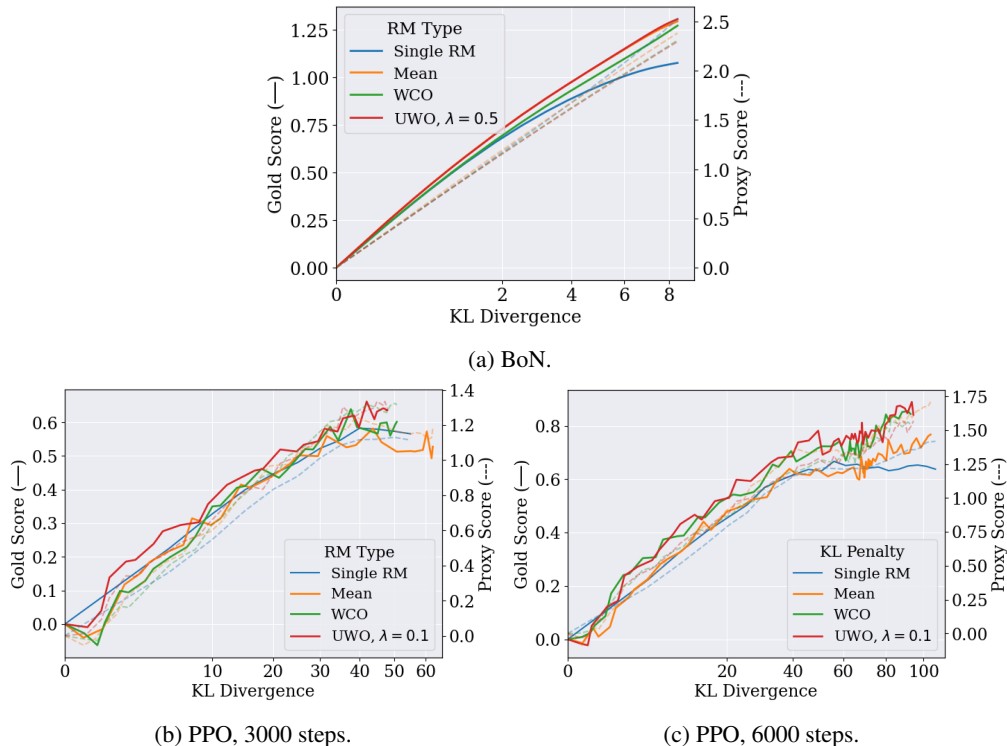

(a) BoN.

(b) PPO, 3000 steps.       (c) PPO, 6000 steps.

Figure 19: BoN and PPO results with the 1.3B reward models to supplement the bar plot in Figure 8. Although all PPO experiments are run for 3000 steps, here we also include an experiment with 6000 steps for the 1.3B reward model, due to the policy being optimized at a slower rate for large reward models. Differences can be observed at higher KL divergence, which is generally reached in the later steps of PPO training. In particular, the difference in performance between the single RMs and ensemble methods is clearer when the policy is trained for longer.

## F.5 Additional No Label Noise Results

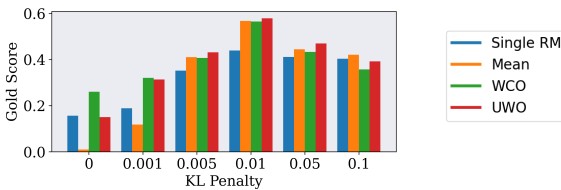

Figure 20: Effectiveness of conservative optimization methods across KL penalty weights in the no label noise case.

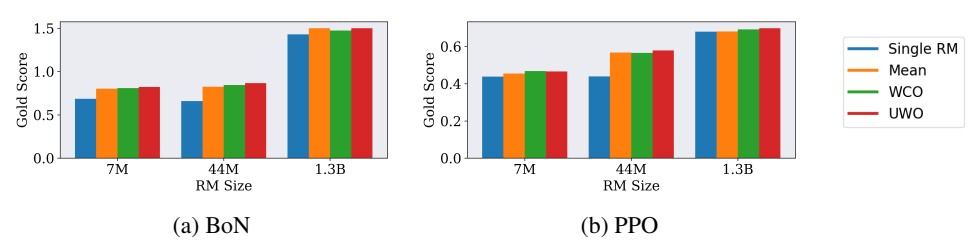

(a) BoN

(b) PPO

Figure 21: Final gold reward model performance achieved by different objectives when optimizing reward models with varying parameter sizes, but trained with the same dataset without label noise.

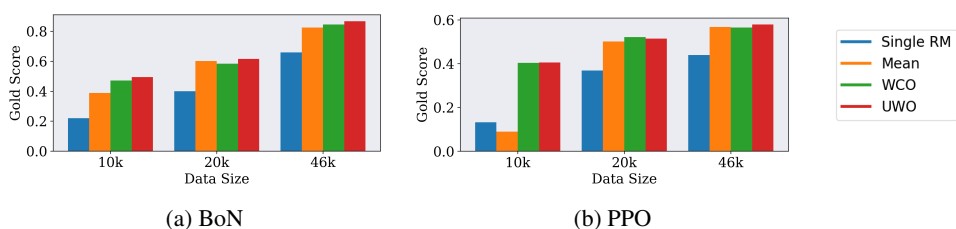

(a) BoN

(b) PPO

Figure 22: Final gold reward model performance achieved by different objectives when optimizing (44M) reward models trained under varying amounts of data. No label noise is used in the data.

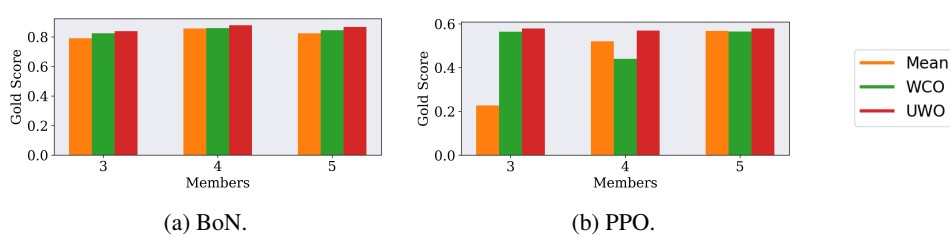

(a) BoN.

(b) PPO.

Figure 23: Impact of the cardinality of the ensemble on the final performance of mean, UWO, and WCO for the no label noise case.

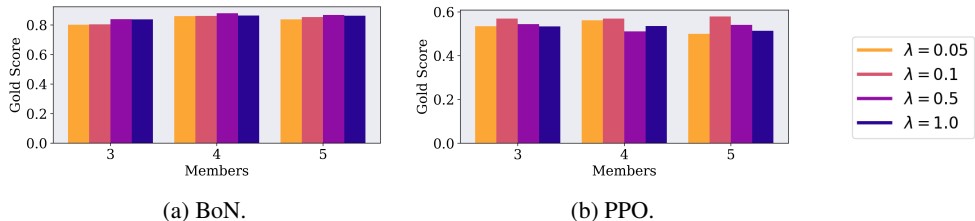

(a) BoN.               (b) PPO.

Figure 24: Impact of uncertainty penalty weight on final performance for different numbers of reward models within the ensemble. We note that there is considerable robustness to the value of the uncertainty penalty. No label noise is included in the data.

### F.6 35% LABEL NOISE RESULTS

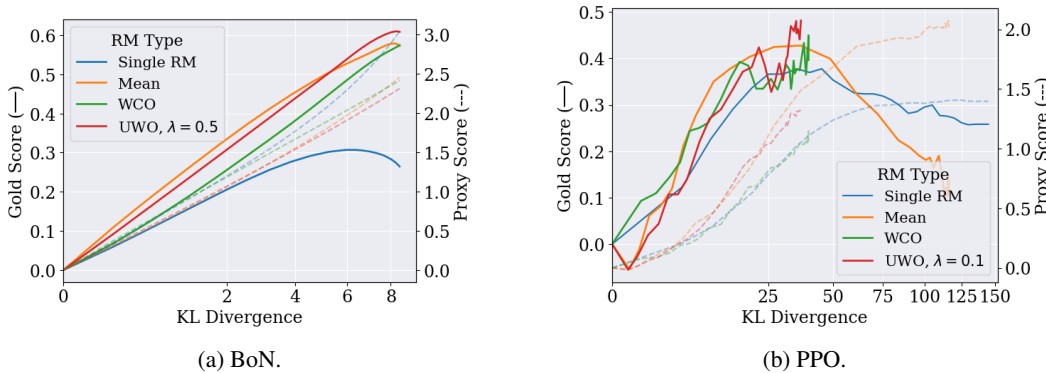

(a) BoN.               (b) PPO.

Figure 25: BoN and PPO results for 44M reward models with KL penalty 0.01, when there is 35% label noise in the reward model training data. Results show that the increase in performance and overoptimization mitigation of ensembles is robust to a different noise level.

### F.7 LONGER PPO RESULTS

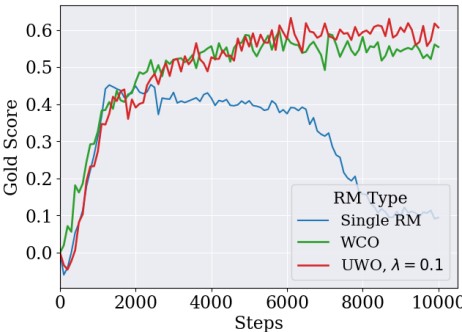

Figure 26: PPO results for 44M reward models with KL penalty 0.01 and 25% label noise. Optimization is extended to 10k steps for this experiment. These results show the stability of ensemble methods even for high levels of optimization. Thus, our ensembling removes the need for early stopping and the risk of overoptimziation. It would seem that no matter how far the model is optimized, the performance will stay close to optimal.

### F.8 FULL UNCROPPED KL DIVERGENCE PPO RESULTS

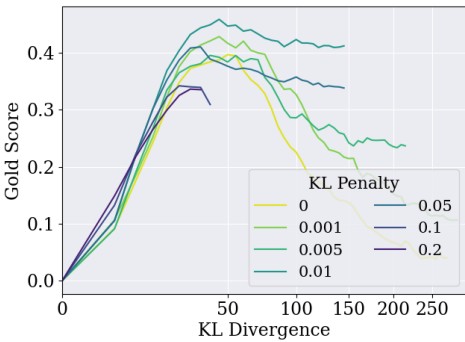

Figure 27: Individual reward model optimization (i.e. no ensemble) for different values of KL penalty in the 25% label noise case. This figure does not crop the x-axis at 150 KL divergence, to illustrate the behavior of Fig. 6 beyond this point. As mentioned in Section 5, we usually stop our optimization curves at 150 KL divergence because the overoptimization trends are already visible, and doing so provides consistency across figures. Moreover, the plot is clearer as it isn't made narrow by more extreme values.

### F.9 WIN-RATE EVALUATION OF ENSEMBLE METHODS

As an additional measure of quality, Tables 6 and 7 show the win-rate of final policies trained with the different ensemble methods, when compared with the single reward models, across multiple reward model scales. These tables correspond to runs shown earlier in Figures 3, 4, 18 and 19. For BoN, we use the unbiased estimator at $n = 12500$ and compare gold scores between each method and the single reward models for each prompt. For PPO, we compare the gold scores of answers generated by the policy after 3000 PPO training steps. Win-rate is calculated by averaging the individual win-rates against each of the five single reward models, with standard deviation also presented. Results for PPO are additionally averaged over the three PPO seeds.

Table 6: Final policy win-rates (in percent) of ensemble methods against single reward models, when using 44M reward models. Standard deviation across the five single reward models is indicated. Models correspond to those shown in Figures 3 and 4.

|                      | BoN | | PPO | |
| --- | --- | --- | --- | --- |
|                      | No noise | 25% noise | No noise | 25% noise |
| Mean                 | $54.9 \pm 0.9$ | $57.8 \pm 1.3$ | $58.1 \pm 3.3$ | $60.2 \pm 3.5$ |
| WCO                  | $57.1 \pm 0.6$ | $58.3 \pm 0.8$ | $59.4 \pm 3.3$ | $62.2 \pm 2.9$ |
| UWO $\lambda = 0.5$  | $57.2 \pm 0.9$ | $58.2 \pm 1.0$ | $60.2 \pm 3.4$ | $63.0 \pm 3.1$ |

Table 7: Final policy win-rates (in percent) of ensemble methods against single reward models, when using smaller (7M) and larger (1.3B) reward models. Standard deviation across the five single reward models is shown. Noise level is fixed to 25%, such that models correspond to those in Figures 18 and 19. The 6000-step final policy is used for PPO in the 1.3B reward model case.

|                      | BoN | | PPO | |
| --- | --- | --- | --- | --- |
|                      | 7M RM | 1.3B RM | 7M RM | 1.3B RM |
| Mean                 | $57.7 \pm 1.4$ | $72.3 \pm 1.0$ | $50.8 \pm 3.4$ | $56.2 \pm 3.5$ |
| WCO                  | $58.8 \pm 1.4$ | $71.2 \pm 1.2$ | $57.9 \pm 3.7$ | $62.9 \pm 2.9$ |
| UWO $\lambda = 0.5$  | $61.3 \pm 1.4$ | $73.8 \pm 0.8$ | $59.0 \pm 3.7$ | $63.1 \pm 3.2$ |

### F.10 QUALITATIVE SAMPLES

Quantitative metrics are used throughout our experiments as they enable the rapid and comprehensive evaluation of various methods. Here we provide a brief insight into the qualitative aspect of the methods discussed in this work. For a given prompt, answers from the final policy ($n = 12500$ for BoN, and 3000 steps for PPO) of each method are provided. Policies from the runs in Figures 3 and 4, which provide the strongest final results, are used.

The main takeaways are as follows. First, the BoN answers are shorter, because policy optimization does not occur in the same way it does for PPO. Second, single reward models show signs of failure and overoptimization. For BoN, this translates to poor answers and even hallucinations in the case of Table 8. For PPO, this also translates to poor answers, which are very long and highly repetitive. These are clear instances of overoptimization, where the reward model seems to have learned to produce long answers regardless of the content, which offers good proxy reward model performance but poor gold model performance. Finally, we note that the conservative optimization methods produce strong qualitative results, even when single reward models are struggling.

Table 8: First set of example answers to an evaluation prompt.

| | |
|---|---|
| Prompt | Find the closest airport to the given location.
Seattle, Washington |
| Single RM (BoN) | The closest airport to the given location is Sea-Tac Airport, with a direct flight from SFO airport in San Francisco, CA to Seattle-Tacoma International Airport available on up to dates. |
| Mean (BoN) | The nearest airport to Seattle, Washington is Sea-Tac International Airport (SEA). It is located a distance of 17.6 miles away and has an estimated time of arrival of around 12:30 PM. |
| WCO (BoN) | The closest airport to the given location is Seattle-Tacoma International Airport (SITE). It is located approximately 10.7 miles away from the given location. |
| UWO (BoN) | The closest airport to the given location is Seattle-Tacoma International Airport. It is located 23.7 miles away, and the estimated time of arrival is 2 hours, 28 minutes. |
| Single RM (PPO) | The closest airport to the given location is Seattle-Tacoma International Airport, located in the Seattle, Washington, region. It is a major airport with a major hub, serving over 15 million passengers each year. It is located just 10 miles from the city center and is connected to the city by the Bunker Hill and Space Needle commuter rail lines. It is the primary airport for the region and is one of the most important hubs in the country. It is a major hub for a variety of airlines, and is a popular destination for travelers from around the world. It is a hub for a variety of domestic and international flights, and is the main hub for the region. It is a major hub for several airlines, and is a popular destination for travelers from around the world. It is a major hub for a variety of domestic and international flights, and is the primary hub for the region. It is a major hub for a variety of airlines, and is a popular destination for travelers from around the world. It is a major hub for a variety of domestic and international flights, and is the main hub for the region. It is a major hub for a variety of airlines, and is a popular destination for travelers from around the world. |

*Continued on the next page*

| Prompt | Find the closest airport to the given location.
Seattle, Washington |
|---|---|
| Mean (PPO) | The closest airport to Seattle is Boeing Field, located in Seattle, Washington. It is served by several airlines, including Alaska Airlines, United Airlines, Delta Air Lines, and American Airlines. It is located approximately 35 miles away from the given location. The airport is also served by several car rental companies, including Hertz, Avis, and Budget. |
| WCO (PPO) | The closest airport to the given location is Seattle-Tacoma International Airport (SEA). It is located approximately 43 miles from the location. The average time to get to the airport is around 2 hours. The cost to get to the airport is around $110 for a single-occupancy car. It is the most popular airport in the area and is served by many airlines. The airport offers a variety of services, including a baggage claim, a car rental counter, and a currency exchange. There are also a number of attractions nearby, including the Space Needle, the Aquarium of the Pacific, and the Seattle Center. The airport is well-served by public transportation, making it easy to get around. |
| UWO (PPO) | The closest airport to the given location is Seattle-Tacoma International Airport (SEA). The airport is located in the Seattle-Tacoma, Washington, area and is served by many domestic and international airlines. It is located about 70 miles east of the city of Seattle and is accessible by car, bus, train, and ferry. The airport has several terminals, including a domestic terminal, an international terminal, and a car rental center. The airport has two runways, allowing for both domestic and international flights. The airport is served by many international carriers, including Alaska, American, Delta, United, and Virgin. It is also home to many domestic carriers, including American, Delta, and United. |

Table 9: Second set of example answers to an evaluation prompt.

| Prompt | Create an analogy to explain cloud computing. |
|---|---|
| Single RM (BoN) | Cloud computing is like the sky; it is a vast, untamed expanse filled with a variety of colorful shapes and textures. Just like clouds move and shift based on the weather, so too does cloud computing. It is constantly evolving and adapting to meet the demands of modern technology. |
| Mean (BoN) | Cloud computing is like a sprawling house, with multiple rooms and floors that each represent a specific type of resource. Each room serves a specific purpose, such as storage, processing, or network access, and each floor has its own features such as storage, computing power, or network connectivity. The cloud provides a central hub where resources can be accessed from anywhere in the house, and enables users to access the resources they need quickly and easily. |

*Continued on the next page*

| Prompt | Create an analogy to explain cloud computing. |
|---|---|
| WCO (BoN) | Cloud computing is like a giant server room in your home, with multiple computers connected together to store, process, and retrieve data. The computers are like small servers in the server room, performing different tasks such as running virtual machines, performing analytics, and performing other tasks. Each computer is connected to the server, and the data stored in the server is retrieved and used by the computers. The data is stored in a database, allowing data to be quickly and easily accessed, updated, and processed. |
| UWO (BoN) | Cloud computing is like a stack of pancakes; each pancake is a machine that processes data and provides computing power. When you want to make pancakes, you simply select the ingredients, put them in a pan, and flip them to make pancakes. Similarly, when you use cloud computing, you select the services you need to make use of, then pay a monthly fee to use them. When you're done with your work, you can shut down the machine and delete the files, or if you need more power, you can simply purchase more power and pay a recurring fee to make use of it. |
| Single RM (PPO) | The analogy is that of a computer being a network of interconnected servers, each with its own processing power and memory, and the cloud as the central hub where the data is stored, allowing the various servers to access the data in a more efficient and cost-effective manner. It is the cloud that allows the various servers to be able to access the same data in the same way, and the data is stored and managed in the cloud, thus allowing the various servers to work together to provide a more efficient and cost-effective solution. It is the cloud that is the backbone of the modern technology and it is the ability to access the data from the various servers, in a more cost-effective and efficient manner, that makes cloud computing the ideal solution for the modern technology. It is the cloud that is the future of technology and it is the ability to access the data from the various servers, in a more cost-effective and efficient manner, that is the foundation of the modern technology. It is the cloud that is the new standard and it is the ability to access the data from the various servers, in a more cost-effective and efficient manner, that is the foundation of the future of technology. It is the cloud that is the future of technology, and it is the ability |
| Mean (PPO) | Cloud computing is like a giant cloud of air that provides all of the computing power needed to run a business. The cloud provides access to the necessary computing resources to enable businesses to remain competitive and stay ahead of the competition. Cloud computing is a cost-effective way to access the computing power needed to run a business, and it can reduce the time it takes to develop and deploy new software. It also allows businesses to scale their operations quickly, with the ability to quickly add or remove resources as needed. Cloud computing is a great way for businesses to remain competitive and stay ahead of the competition. |

| Prompt | Create an analogy to explain cloud computing. |
|---|---|
| WCO (PPO) | Cloud computing is like a huge, open room with many different walls and floors. The walls provide security and privacy, while the floors provide resources and data access. The ceiling is the sky, and the sky is the sky, providing a vast amount of resources, power, and opportunities. The cloud is a room full of resources, and the resources are the people, providing a diverse community of expertise and resources. The walls, floors, and sky are all important, and together they provide an invaluable resource. This analogy helps to explain the importance of cloud computing, and how it can be beneficial to organizations of all sizes. It shows the power of the cloud and the opportunities it provides, allowing organizations to scale quickly and efficiently. It also illustrates the importance of providing a secure, reliable, and cost-effective infrastructure. |
| UWO (PPO) | Cloud computing is like a modern-day library, where data is stored and processed on a centralized server. The data is accessed and manipulated from any device connected to the internet, allowing for faster and more efficient services. The data is stored in the cloud and is accessible from any device, allowing for the user to access the data quickly and efficiently. The cloud computing platform is easy to use, provides the user with the flexibility to access the data quickly and easily, and is cost effective. The cloud computing platform provides the user with the ability to access the data quickly and easily, allowing them to make decisions based on the data. The cloud computing platform also provides the user with the flexibility to access the data quickly and easily, allowing them to make decisions based on the data. This allows the cloud computing platform to be a cost effective solution for the user, while providing them with the ability to access the data quickly and easily. |

