# OpenReview forum: "Reward Model Ensembles Help Mitigate Overoptimization"
_ICLR.cc/2024/Conference — ICLR 2024 poster_

### Official Review · Reviewer_PF7E · 2023-10-28

**Soundness:** 3 good
**Presentation:** 4 excellent
**Contribution:** 3 good
**Rating:** 6
**Confidence:** 3

**Summary:**

The authors tackles the over-optimization issue in RLHF with reward model ensembles. This is achieved by training multiple reward models, each with identical data but different random seeds. These reward models are then used for ensemble-based optimization objectives, including worst-case optimization (WCO) and uncertainty-weighted optimization (UWO) for best-of-n sampling (BoN) and PPO (proximal policy optimization). Combined with a 25% label noise to mirror real-world settings, the authors show UWO and WCO can effectively mitigate overoptimization.

**Strengths:**

* **Simple yet elegant approach**: the authors only used different random seeds to train the reward function, yet it can significantly improve gold score performance, especially with BON.
* **Reproduced overoptimization with OSS (open-source software) models**: the authors are the first to study and reproduce RM overoptimization with open-source models.
* **Experiments across multiple model sizes / data sizes**: the authors conducted a comprehensive analysis, which offers insights.

**Weaknesses:**

**Lack of account for random seeds**: the results do not look smooth for Figure 4 / Figure 8. The results could be improved by running with at least 3 random seeds and recording the error bars.

**Questions:**

> we use the complete dataset of 46, 000 prompts for reward model training and train all the reward models for 5 epochs.

>We use all the available training data for the training of every reward model as
training on lesser data results in higher validation loss and poorer performance (Lakshminarayanan
et al., 2017). Unless stated otherwise, we train an ensemble consisting of five reward models.

Why train for 5 epochs? What is the training and the validation accuracy of the reward model? In Figure 15, can you show me the epoch as the x-axis? Steps can be confusing as it is related to the batch size / gradient accumulation / how you log.

> we only evaluate BoN for a maximum of nmax = 12, 500 samples , which roughly equals 8.4 nats of KL

Why nmax? Does this mean for some of them he use less than 12500 samples?

> we train for 3000 PPO steps.

How many episodes?


> Figure 3

Why do you have two y-axis? Do the proxy RM and the gold RM have different RM scales? How are these scales defined?


> Figure 6 and 7

Figure 6 and 7 seem contradictory? KL penalty = 0 gets 0.15 gold score, but in figure 7 KL penalty = 0 gets 0.03 gold score?

> Figure 8

Why does PPO underperform BoN in 1.3B setting according to Gold Score? Gao et al (2023) show 1.2B PPO outperforms 1.2B BoN

> Appendix C

How does these two KL distance calculation Gold RM performance?


> We train a fixed number of proxy reward models using
identical data and hyperparameters but with different random seeds

But I assume data shuffling is different? There are really multiple random seeds:

* query dataset seed
* reward model seed
* policy model seed


> Figure 4
Why is single RM experimented with KL=150 but not the other types?

---

> ### Author Response · Authors · 2023-11-17
>
> Thank you for your review. We’re glad you found the approach simple and elegant, and the experiments comprehensive. We will now address your questions in detail.
>
> We agree that having multiple PPO seeds will improve the results in our paper, thank you for the suggestion. We have started running this and will add the results soon. We didn’t run these experiments previously because we had observed very low variance with different PPO seeds in initial experiments, and the computational cost of running multiple seeds is high.
>
> > Why train for 5 epochs? What is the training and the validation accuracy of the reward model? In Figure 15, can you show me the epoch as the x-axis? Steps can be confusing as it is related to the batch size / gradient accumulation / how you log.
>
> We use 5 epochs because this is the point at which validation loss plateaus, without overfitting. This can be seen in Figure 15. While Figure 15 shows steps and not epochs, the run is for 5 epochs, such that an epoch occurs about every 359 steps. We have updated the caption of the figure to reflect this. Thank you for the suggestion to improve the clarity of the paper.
>
> With regards to RM accuracy, for example in Figure 15 the validation accuracy is around 67%. RM accuracy generally tends to be between 60-75%. We have added this information in Section 4.3, under “Proxy Reward Model Training”.
>
> > Why nmax? Does this mean for some of them he use less than 12500 samples?
>
> For BoN we always evaluate for 12500 samples for the largest values of n. The nmax comes from the unbiased estimator used (see Appendix I in [1] for greater detail), and refers to the greatest value of n we evaluate BoN for. For lower values of n, we of course use fewer samples.
>
> > we train for 3000 PPO steps.How many episodes?
>
> Each PPO step is 32 episodes, so 3000 steps is 96,000 episodes, which is just about 5 epochs over the unlabeled RL data.
>
> > Figure 3. Why do you have two y-axis? Do the proxy RM and the gold RM have different RM scales? How are these scales defined?
>
> In e.g. Figure 3 we have two y-axis because although the proxy and gold scores have roughly the same scale (i.e. the RMs are normalized to 0 mean and unit variance), the proxy scores grow much higher. We therefore add a second y-axis to avoid dwarfing the gold scores and making them harder to read.
>
> > Figure 6 and 7 seem contradictory? KL penalty = 0 gets 0.15 gold score, but in figure 7 KL penalty = 0 gets 0.03 gold score?
>
> While most PPO runs terminate before or around when they reach a KL divergence of 150, some significantly exceed this, particularly for single RMs and when using low KL penalties. In order to keep the plots as legible as possible, we truncate runs that exceed this KL divergence when plotting the optimization curves. Past this point, the overoptimization trend has already been observed and the x-axis risks growing long and making all other runs less clear. Moreover, this enables us to have consistent plots with equivalent x-axes. We have updated the manuscript to include this information in the introduction to Section 5. This explains why runs with very low KL values have final gold scores which are lower in the bar plots (Figure 7) than on the optimization curves (Figure 6). While the visualization curves are truncated at a KL divergence of 150 (by which point the presence of overoptimization has been confirmed and performance has already declined), the bar plots record the final values at the very end of the run. We also add an equivalent of Figure 7 without the truncation in Appendix F.8 to illustrate this.
>
> We continue answering in the following comment, due to character count limitations.

---

> > ### Author Response · Authors · 2023-11-17
> >
> > > Figure 8. Why does PPO underperform BoN in 1.3B setting according to Gold Score? Gao et al (2023) show 1.2B PPO outperforms 1.2B BoN
> >
> > In Gao et al. [2], page 3, Figure 1, the PPO 1.2B model reaches a peak performance of about 1.0, while the BoN 1.2B reaches over 1.15. Which part of Gao et al do you think shows 1.2B PPO outperforming 1.2B BoN? In addition, Gao et al did not open-source their implementation and used closed-sourced models, so it is difficult to know exactly what the differences might be. We show some basic reproductions of Gao et al.’s results in Appendix E.
> >
> > > Appendix C. How does these two KL distance calculation Gold RM performance?
> >
> > We are not perfectly clear on what you mean in this question. If you meant which KL is used in our plots of gold RM performance, we use the second type of KL distance (eqn 10) for PPO. Additionally the choice of equation does not affect gold RM performance, it just allows us to plot KL without negative values and provides a clearer visualization.
> >
> > > But I assume data shuffling is different? There are really multiple random seeds
> >
> > Indeed data shuffling changes with the random seed. For RM training, the random seed affects both the data shuffling and head initialization, and for each run both are randomised based on the seed. PPO also has its own seed for training. The results for PPO currently in the paper are only for one seed, but we are running additional seeds for PPO training to ensure our results are robust.
> >
> > > Figure 4 Why is single RM experimented with KL=150 but not the other types?
> >
> > In Figure 4 the single RM is the only one which reaches a KL divergence of 150, the ensemble methods never reach such a high KL distance. It appears that the conservative optimization methods limit the KL divergence of the optimized policy.
> >
> >
> > We hope we have addressed all your concerns and questions, and if so, we ask if you would consider raising your score?
> >
> > [1] Nakano et al. WebGPT: Browser-assisted question-answering with human feedback.
> >
> > [2] Gao et al. Scaling Laws for reward model overoptimization.

---

> > > ### Comment · Reviewer_PF7E · 2023-11-22
> > > **Reply**
> > >
> > > I think the authors for their detailed replies which have addressed most of my concerns.
> > >
> > > > In Gao et al. [2], page 3, Figure 1, the PPO 1.2B model reaches a peak performance of about 1.0, while the BoN 1.2B reaches over 1.15. Which part of Gao et al do you think shows 1.2B PPO outperforming 1.2B BoN?
> > >
> > > I apologize — I must have seen the wrong figures.
> > >
> > > > Appendix C. How does these two KL distance calculation Gold RM performance?
> > >
> > > I meant have you tried using tried running training with these two KL estimators, respectively, and observing how they affect performance. http://joschu.net/blog/kl-approx.html also proposes a third KL estimator.

---

> > > > ### Author Response · Authors · 2023-11-22
> > > >
> > > > Thank you for your response, we are happy to hear we have addressed most of your concerns.
> > > >
> > > > We agree it would be interesting to look at how different KL estimators affect PPO training and performance, but we believe this is out of the scope of this work. In our experiments, we follow standard practice and have not modified the PPO KL estimator.
> > > >
> > > > Schulman’s KL divergence blog does indeed propose a third estimator, but this estimator is unstable - when the KL divergence grows large as training progresses, the value returned by the estimator explodes due to its exponential term.

---

### Official Review · Reviewer_249u · 2023-10-31

**Soundness:** 3 good
**Presentation:** 4 excellent
**Contribution:** 3 good
**Rating:** 8
**Confidence:** 4

**Summary:**

This paper addresses the problem of reward model optimization in training of LLM chatbots: when optimizing a learned reward model initially improves the chatbot's performance under "true" human preferences but eventually the performance decreases as the learned reward ceases to be a good proxy. The authors propose training an ensemble of reward models instead of a single one to mitigate this. The ensemble is trained from different random seeds and then the optimization target is either the mean of the reward models' outputs, the minimum (WCO), or the mean minus a variance penalty (UWO). A number of experiments are performed on best-of-n (BoN) and PPO optimization of reward models using these optimization targets and they are compared to using the output of a single reward model. The results suggest that using an ensemble of reward models

**Strengths:**

The paper addresses a very important problem—one of the main bottlenecks to improving RLHF training and safety of LLM-based chatbots is making reward models more robust. The solution technique is not particularly novel, as pessimistic optimization using an ensemble has been widely used in model-based RL, offline RL, preference learning, etc. However, I don't know of prior work that has specifically evaluated this technique for RLHF on LLM-based chatbots. Thus, I view the primary contribution of this paper as a systematic empirical study of applying this existing technique to a new problem. I think this is an important contribution but it does mean the experiments should be carefully executed.

In general, the experiments do seem to be well-done, exploring the effects of using the different ensemble-based optimization objectives in a variety of settings. Using WCO and UWO seem to pretty consistently help; while they don't allow for completely removing KL regularization in PPO, they do allow PPO to reach a higher final gold reward.

The writing is in general quite clear and the paper is easy to follow.

**Weaknesses:**

In terms of the experiments, one weakness is that the PPO experiments seem to be mostly done with a single random seed, while due to the high noise in RL training it is best to use a few random seeds (see https://arxiv.org/abs/2108.13264, https://arxiv.org/abs/2304.01315).

Another weakness of the results is that it's hard to know how to interpret the gold reward. How much better is an LLM with an average gold reward increase of 0.5 vs. 0.4? AlpacaFarm and others use a win-rate which is more interpretable—it might be helpful to report that for your results as well so that it's easier to understand exactly how much better using an ensemble is.

The way the PPO results are presented analogously to Gao et al. also feels a bit strange. According to Figures 4 and 5, the ensemble-based optimization objectives only outperform a single reward model at the very end of PPO training. I'm not sure why it's important to see all the intermediate KL and reward values when early-stopping is not generally used with RLHF PPO training. I found Figure 7 to be the best comparison of the different optimization objectives. It would be particularly interesting to see a variation of Figure 7 where each the sqrt(KL) and gold reward is plotted for each combination of (optimization objective, KL coefficient) since the KL coefficient is the actual "knob" used to adjust how much optimization is occurring in practice (not the amount of training time).

Here is one paper that should probably be included in the related work as they also evaluate using ensembles of LLM-based reward models: https://arxiv.org/pdf/2203.07472.pdf. Their paper is focused on active learning, not reward optimization, so it's not directly comparable, but it does relate to your conjecture that "uncertainty estimates from the ensemble may also help improve sample efficiency of human feedback in this setup" on page 9.

Small issues/typos:
 * Bottom of page 5: in two places you say "for PPO, WCO, and UWO" but the second comma should be removed

**Questions:**

Related to the weaknesses above:
 * What are the results of running PPO across 3-5 random seeds for the various optimization objectives? How much variance is there across seeds?
 * How much does the gold reward increase from using an ensemble correspond to in terms of win rate or other measures of quality?

---

> ### Author Response · Authors · 2023-11-17
>
> Thank you for your review. We’re glad you find the problem of overoptimization an important one, and that you think the experiments are well done. We now respond to your comments in detail.
>
> > one weakness is that the PPO experiments seem to be mostly done with a single random seed, while due to the high noise in RL training it is best to use a few random seeds
>
> We have started to run multiple PPO seeds and will add these results soon for e.g. Figure 4. The reason for not having included these from the start is because we had observed very low variance when using different PPO seeds in initial experiments and the computational cost is very high.
>
> > Another weakness of the results is that it's hard to know how to interpret the gold reward. How much better is an LLM with an average gold reward increase of 0.5 vs. 0.4?
>
> Thank you for the suggestion of providing a more interpretable measure of quality. We will perform a win-rate comparison between single RMs and the conservative optimization objectives to offer an additional measure of quality, and add that to the paper once it is complete. The samples in Appendix F5 can also serve as a qualitative assessment of the difference in quality.
>
> > I'm not sure why it's important to see all the intermediate KL and reward values when early-stopping is not generally used with RLHF PPO training.
>
> Though it seems that conservative optimization objectives only outperform single RMs towards the end, it is helpful to provide the full intermediate rewards to better show how overoptimization occurs, and that the reward does indeed rise, plateau, and then fall for single RMs. While providing only the final reward gives a good indication of final performance, it doesn’t illustrate the presence of overoptimization, or the mitigation of this behaviour, as well.
>
> > I found Figure 7 to be the best comparison of the different optimization objectives. It would be particularly interesting to see a variation of Figure 7 where each the sqrt(KL) and gold reward is plotted for each combination of (optimization objective, KL coefficient) since the KL coefficient is the actual "knob" used to adjust how much optimization is occurring in practice (not the amount of training time).
>
> We are not entirely clear on your suggestion for a variation of Figure 7, would you be able to clarify your suggestion and the motivation behind it? Figure 6 shows results for using different KL coefficients. In particular this demonstrates that both the KL penalty term and our conservative optimization are important, because KL penalty alone is either insufficient to prevent overoptimization, or performs significantly worse for large penalties.
>
> > Here is one paper that should probably be included in the related work as they also evaluate using ensembles of LLM-based reward models: https://arxiv.org/pdf/2203.07472.pdf.
>
> Thanks for pointing out this paper. We have included a mention and reference to it in Related Works within the main text.
>
>
> We hope we have addressed all of your concerns, and that you will consider raising your rating of the paper, but please let us know if you have any remaining questions.

---

> > ### Comment · Reviewer_249u · 2023-11-21
> > **Response to authors**
> >
> > Thank you for addressing the points in my review. I think including results for multiple seeds and the win-rate metrics will strengthen the empirical results of the paper. Regarding Figures 6 and 7, I believe I understand these figures better now and I don't think it's necessary to change them.
> >
> > I understand that there is a short timeline for additional experiments, but ideally it would be great to see either the win-rate or multiple seeds results during the discussion period. I'm happy to raise my score if either of these can be added to the paper.

---

> ### Author Response · Authors · 2023-11-22
>
> Thank you for your response. We agree that these additional experiments are valuable additions to the paper. As such we have:
> - Added runs with multiple PPO seeds in Figures 4a and 4b. Both the mean and standard deviation across runs are shown.
> - Included win-rate results to compare the performance of ensemble methods to those of single reward models in Appendix F9. We show these for both BoN and PPO for reward models of size 44M and plan on adding other model sizes for the camera-ready version.
>
> We hope to have addressed your concerns and requests, and greatly appreciate you considering raising your score in response to our additions provided they meet your expectations.

---

> > ### Comment · Reviewer_249u · 2023-12-01
> > **Thank you for the additional results**
> >
> > I appreciate the authors' hard work to include these additional results and I have raised my score accordingly.

---

### Official Review · Reviewer_mVSG · 2023-11-10

**Soundness:** 3 good
**Presentation:** 3 good
**Contribution:** 3 good
**Rating:** 6
**Confidence:** 4

**Summary:**

This paper investigates the issue of reward model overoptimization in Reinforcement Learning from Human Feedback (RLHF), a technique used to fine-tune large language models. Overoptimization occurs when learned reward models, imperfect representations of true human preferences, lead to undesirable behavior. This paper builds on previous work by Gao et al. (2023) who demonstrated the persistence of overoptimization even with large “gold” reward models and extensive training data. The paper explores the effectiveness of ensemble-based conservative optimization (WCO and UWO) in mitigating overoptimization when using BoN sampling and PPO optimization methods. Additionally, the research extends the previous setup by introducing 25% label noise to better simulate real-world conditions. The findings reveal that conservative optimization significantly reduces overoptimization, improving performance by up to 70% for BoN sampling. For PPO, ensemble-based conservative optimization outperforms single reward model optimization and minimizes overoptimization when combined with a KL penalty, without sacrificing performance. In summary, the authors show that ensemble-based conservative optimization presents a promising approach for addressing the challenge of overoptimization in RLHF, leading to more robust and reliable fine-tuning of large language models.

**Strengths:**

1. This paper provides extensive empirical evidence that suggests that ensemble-based methods can improve robustness of RMs and reduces overoptimization, which makes the claims of the paper well-supported.

2. The paper studies the important problem in RLHF, i.e. reward overoptimization, and presents various methods that clearly mitigate such a challenge. I think the paper is of value to the RLHF community.

**Weaknesses:**

1. While the empirical results are quite comprehensive in the paper, the model size seems a bit small with the biggest RM being 1.3B. Given Figure 8, it seems that the gain of the ensemble-based methods diminishes as the model size increases. It would important to investigate if ensemble-based methods have little gain with even bigger models, which are more commonly used by users.

2. From Figure 9, it seems that with bigger dataset size (46K), ensemble-based methods are not that much better than the single RM. Is it expected that with even more data, which in practice is usually true, the difference between ensemble-based methods and single RM will further shrink? It would be helpful to further study this perspective.

**Questions:**

Please clarify and investigate the two questions presented in the weakness section.

---

> ### Author Response · Authors · 2023-11-17
>
> Thank you for your review. We agree that overoptimization is an important problem and we’re glad you found the results encouraging and the conclusions well-supported.
>
> With respect to reward model size and data size, we use the largest RM possible in our setup and with our computing resources. While we agree it would be interesting to conduct experiments using larger models, we believe the current experiments are sufficient evidence that the ensemble and conservative optimisation techniques are useful for mitigating overoptimization, and will continue to be at larger model sizes, for several reasons:
> 1. While the gain in performance seems to decline a little in proportion when increasing the reward model size, the performance gain is still very much present and does not show signs of completely disappearing. For BoN sampling, in Figures 8a and 9a, the gap between single RM performance and ensemble-based methods performance remains approximately similar in size across different scales - this suggests that ensembles likely follow a similar scaling law for overoptimization as a single RM, but shifted upwards.
> 2. While RM sizes can be increased and larger RMs tend to see less overoptimization, this performance increase eventually hits diminishing returns, as seen in Figure 1 of Gao et al. [1].
> 3. Further, we note that if you look at Figure 19, which shows the full training curve for 1.3B RM - it seems that for both BoN and PPO - while the training appears to have converged for single RM, it does not appear to have converged for WCO and UWO. Moreover, the single RM has not had a chance to properly overoptimize and see a decrease in performance. These claims are backed up by the new results in Appendix F.7, which show that our PPO ensembles can be further trained - resulting in higher performance for ensemble methods, and lower final performance for the single RMs, which overoptimize. This will be especially true for larger RMs, which are optimized at a slower rate.
> Within our setup, it is not possible to further optimize BoN due to computational constraints. However, for PPO, we plan on adding results of optimizing 1.3B RM with an increased training budget of 6000 steps. This is however going to be a very expensive experiment (we expect this to take ~72 hours to finish using the computing resources available to us), hence, we can only give weak commitment that we will be able to show this result to the reviewer within the rebuttal period. However, we do commit to including this result in the camera ready version if the paper gets accepted.
> 4. The new results in Appendix F.7 also illustrate another advantage of the ensemble methods over larger RM sizes: stability and robustness to training time. These models can be trained for a long time without fear of seeing a decrease in performance due to overoptimization. This is not true even of very large RMs, which are known to suffer from overoptimization.
>
>
> We hope we have addressed all your concerns and questions about the paper, and that you will consider raising your score, but please let us know if you have any remaining questions.
>
> [1] Gao et al. Scaling Laws for reward model overoptimization

---

> > ### Author Response · Authors · 2023-11-22
> >
> > To follow up on our previous comment, we have added results for the longer 6000-step PPO run of the 1.3B reward model in Figure 19. As a result, PPO trained with WCO or UWO now peaks at a gold reward model score of 0.84; they were peaking at 0.6 and 0.64 respectively previously (when trained for 3000 steps) while the gold-reward model score for PPO policy trained with single reward model only changed from 0.56 to 0.64. To make it easier for reviewers to compare the gains achieved by further optimization, we have kept both the original 3000-step results (figure 19b) and added the new 6000-step results (figure 19c). We have also updated Figure 8b to include this new result, which shows an increase in performance of more than 30% of the ensemble methods compared to previous values. Both WCO and UWO now outperform the single reward models by over 30% by the end of the 6000 steps. We believe this additional result should allay your concerns regarding diminishing returns from using an ensemble.
> >
> > We would also like to reiterate our mention of Appendix F.7 in our previous comment to the reviewer as well. Along with this new result, these show that with our method, a practitioner does not have to worry about tuning the number of training steps, but can freely choose a large number of training steps without worrying about the dangers of overoptimization.
> >
> > We hope our previous response to your questions, along with this new result has addressed your questions and concerns, and if so ask if you would consider raising your score.

---

### Official Review · Reviewer_83ty · 2023-11-10

**Soundness:** 3 good
**Presentation:** 2 fair
**Contribution:** 3 good
**Rating:** 6
**Confidence:** 3

**Summary:**

This paper studies the overoptimization in the reward model for the RLHF problem and applies several methods to reduce the overoptimization in RLHF. Overall the reviewer believes this submission is above the acceptance bar, but it has some writing issues that can be further improved.

**Strengths:**

1. The experimental results look promising.
2. The overoptimization problem is a novel and important problem to the LLM community.

**Weaknesses:**

1. [Major] There are some writing and presentation issues in the manuscripts. While this manuscript extensively refers to [1] in the writing, the reviewer would recommend the authors update the paper so that the current submission does not require the readers to read [1] to understand the submission thoroughly. See detailed comments in [Questions].
2. [Minor] For paragraph Supervied Fine-tuning in Section 4.3, the hyperlink `(see Section 4.1 for details)` seems to be broken. In addition, the reviewer cannot find any more details in section 4.1 for splitting the AlpacaFarm dataset, perhaps the authors can clarify this?

**Questions:**

1. What does the KL divergence in the x-axis in Figure 2 mean? The reviewer understands [1] also contains similar figures that study the gold score against the KL divergence, and the reviewer believes the authors are using a similar setting. However, the reviewer is unclear which KL divergence is Figure 2 referring to. The authors could either update the caption of Figure 2 or provide a better description of the x-axis to improve the readability.
2. Figure 3 and Figure 4 also have the same issue discussed – perhaps the authors can clarify what is the KL divergence in both figures and why they scale differently. E.g., in Figure 3 the x-axis is from 0-8, while in Figure 4, the x-axis is from 0-150. The reviewer understands that [1] has similar figures, but it will be better if the authors clarify this, so that other readers do not need to refer back to [1] for understanding this submission. Similarly in Figure 5 - 7.

[1] Gao, Leo, John Schulman, and Jacob Hilton. "Scaling laws for rew

---

> ### Author Response · Authors · 2023-11-17
>
> Thank you for your review, and for your positive comments about the experimental results and problem setting.
>
> We thank the reviewer for their suggestions to improve the completeness and clarity of the paper with respect to Gao et al. [1]. Upon your suggestion, we have added further information on KL divergence calculation, including additional explanation of the KL in the captions of the first few plots. We explain this in detail below. We hope this has addressed your concern, but if you feel there are additional details that we should mention in our paper to increase completeness, please do point them out and we would be happy to oblige.
>
> To answer your questions:
> 1. The KL divergence in the x-axes of most figures is the KL distance between the initial policy and the policy being optimized. For BoN, this distance is calculated as per equation (1), and for PPO as per the method in Appendix C. These are in line with Gao et al. [1]. Figure 2 was intended as a generic and pedagogical example of overoptimization, hence the initial lack of detail in describing the KL divergence. We have updated the caption for this Figure to be clearer, as well as added some information regarding the meaning and calculation of the KL divergence for PPO in Section 2.2.
> 2. For Figures 3 and 4, the difference in x-axis is due to the different methods (BoN and PPO) calculating and consuming KL divergence differently. Gao et al. [1] show that PPO seems to consume KL much more than BoN when calculating KL divergence in the way described in the above sections. In addition to the clarification added in the previous point, we further clarify the significance of the KL divergence as well as the difference in scale between BoN and PPO in the captions of Figure 3 and 4.
> 3. For the hyperlink to section 4.1, it seems to work for us? The section linked provides more information about the dataset we use as a whole. We have also added additional details such as the dataset splits in Appendix D.2, and reference this in Sections 4.1 and 4.3 of the main paper for clarity. Even further details can be found in the AlpacaFarm paper [2].
>
>
> We hope this has addressed all your comments and concerns. If so, would you consider increasing your rating of the paper?
>
> [1] Gao et al. Scaling Laws for reward model overoptimization.
>
> [2] Dubois et al. Alpacafarm: A simulation framework for methods that learn from human feedback.

---

> > ### Comment · Reviewer_83ty · 2023-11-18
> >
> > The reviewer would like to thank the authors for updating the paper promptly. All of my concerns are addressed, thank you!

---

> > > ### Author Response · Authors · 2023-11-20
> > >
> > > Thank you for your rapid reply, we are glad to hear we have addressed all of your concerns. Would you consider raising your rating of the paper?

---

> > > > ### Comment · Reviewer_83ty · 2023-11-21
> > > >
> > > > Dear authors, thanks for the follow-up. While the reviewer's questions/concerns are indeed addressed, the reviewer does not believe this will significantly improve the quality of the paper. Hence the reviewer would like to maintain the recommendation.

---

### Author Response · Authors · 2023-11-17

We thank all the reviewers for their thorough and informative reviews. Here is a general comment addressing updates to the paper. Changes in the uploaded revision are written in blue text in order to facilitate review.
- Firstly, we would like to point out a small mistake in the original manuscript. In all the bar plots i.e. Figures 7,8,9 and 11, the legend mistakenly swaps the labels for WCO and UWO. This mistake has been rectified in the revised pdf. Fortunately, as both WCO and UWO perform similarly, this does not impact our conclusions and the main message of the paper still stands: ensemble-based conservative optimization helps prevent overoptimization and improves performance of RLHF.
- Additionally, we include several new results at various noise levels that further support our conclusion that using ensembles helps prevent overoptimization and improve performance in RLHF.
    - We add equivalents for Figures 7, 8, 9, 11, and 12 without the use of label noise in Appendix F.5. By doing so we extensively show that even in this initial setup proposed by Gao et al. [1], our results and conclusions hold.
    - We add optimization results for every method in a setup with 35% label noise in Appendix F.6. We plan to also add results for 15% label noise, and these will demonstrate the robustness of our results (higher performance, less overoptimization) to different noise levels.
    - We add PPO optimization results for 44M reward models for prolonged runs of 10,000 steps in Appendix F.7. These show the stability of our ensemble methods and demonstrate that users can freely optimize the model without fear of overoptimization.

Furthermore, we are currently working on adding the following new results:
- As suggested by reviewers 249U and PF7E, we are running experiments with multiple PPO seeds for all methods (Figure 4).
- As suggested by reviewer 249U, we are planning on adding results showing win-rate of different (final) policies against the (final) policy trained with a single RM.
- In link to a discussion prompted by reviewer mSVG, we will run a longer experiment of 6000 steps for the 1.3B RM to study the evolution of ensemble performance when this model is optimized further. We hope to possibly show this result by the end of the rebuttal period, but otherwise commit to releasing it with the camera ready version, should the paper be accepted.

[1] Gao et al. Scaling Laws for reward model overoptimization.

Edit: better formatting of bullet points.

---

### Author Response · Authors · 2023-11-22
**Addition of new results in response to reviewer feedback**

We again would like to thank the reviewers for their time and reviews. This general comment comes with the latest revision to the manuscript, in which we make several new additions, as promised in our last response. Changes to the manuscript are made in blue for emphasis.
- [Requested by reviewers 249U and PF7E] We ran PPO for multiple random seeds (total of 3 PPO seeds), and show these results, including mean and standard deviation, in Figures 4a and 4b.
- [Requested by reviewer 249U] We have added win-rate for policies trained using ensemble methods against policies trained using single reward models as an additional measure of quality. This is provided in Appendix F.9 for both BoN and PPO.
- We ran PPO with a longer training budget (6000 steps) for the 1.3B reward model, to illustrate the advantage of ensemble methods over single reward models when the model is optimized further. Results are shown in Figure 19. Because policy optimization is slower when optimizing with a large reward model, this difference in performance was previously less apparent. We also update Figure 8b accordingly.

We thank the reviewers for their feedback during this review process, and kindly ask them to consider raising their ratings in light of our responses and new additions.

---

### Meta-Review · Area_Chair_og7W · 2023-12-09

**Metareview:**

This paper proposes an approach for optimizing conservative estimates of rewards during RLHF. They model an ensemble of reward models and use uncertainty-weighted optimization or worst case optimization with PPO and BoN. Overall, the results in the paper are promsing and exhibit favorable scaling (though arguably, scaling is only done till 1.3B params). The reviewers liked this paper and the paper has improved substantially over the course of the discussion period. Hence, I recommend acceptance.

**Justification For Why Not Higher Score:**

I think the paper is at the borderline of a spotlight or a poster -- I would imagine that showing the validity of the approach at a slightly larger scale (>= 3B), would make it a spotlight.

**Justification For Why Not Lower Score:**

The paper has convincing and strong results, well thought out idea and motivation, and was well received by reviewers.

---

### Decision · Program_Chairs · 2024-01-16

Accept (poster)